# Long-Term Use of Amoxicillin Is Associated with Changes in Gene Expression and DNA Methylation in Patients with Low Back Pain and Modic Changes

**DOI:** 10.3390/antibiotics12071217

**Published:** 2023-07-21

**Authors:** Maria Dehli Vigeland, Siri Tennebø Flåm, Magnus Dehli Vigeland, Ansgar Espeland, Manuela Zucknick, Monica Wigemyr, Lars Christian Haugli Bråten, Elisabeth Gjefsen, John-Anker Zwart, Kjersti Storheim, Linda Margareth Pedersen, Kaja Selmer, Benedicte Alexandra Lie, Kristina Gervin

**Affiliations:** 1Department of Research, Innovation and Education, Division of Clinical Neuroscience, Oslo University Hospital, 0450 Oslo, Norway; 2Faculty of Medicine, University of Oslo, 0313 Oslo, Norway; 3Department of Medical Genetics, Oslo University Hospital and University of Oslo, 0450 Oslo, Norway; 4Department of Radiology, Haukeland University Hospital, 5021 Bergen, Norway; 5Department of Clinical Medicine, University of Bergen, 5020 Bergen, Norway; 6Oslo Centre for Biostatistics and Epidemiology, University of Oslo, 0313 Oslo, Norway; 7Department of Physiotherapy, Oslo Metropolitan University, 0167 Oslo, Norway; 8National Center for Epilepsy, Oslo University Hospital, 1337 Sandvika, Norway; 9Pharmacoepidemiology and Drug Safety Research Group, Department of Pharmacy, School of Pharmacy, University of Oslo, 0313 Oslo, Norway

**Keywords:** antibiotics, amoxicillin, gene expression, DNA methylation, immunoglobulin

## Abstract

Long-term antibiotics are prescribed for a variety of medical conditions, recently including low back pain with Modic changes. The molecular impact of such treatment is unknown. We conducted longitudinal transcriptome and epigenome analyses in patients (*n* = 100) receiving amoxicillin treatment or placebo for 100 days in the Antibiotics in Modic Changes (AIM) study. Gene expression and DNA methylation were investigated at a genome-wide level at screening, after 100 days of treatment, and at one-year follow-up. We identified intra-individual longitudinal changes in gene expression and DNA methylation in patients receiving amoxicillin, while few changes were observed in patients receiving placebo. After 100 days of amoxicillin treatment, 28 genes were significantly differentially expressed, including the downregulation of 19 immunoglobulin genes. At one-year follow-up, the expression levels were still not completely restored. The significant changes in DNA methylation (*n* = 4548 CpGs) were mainly increased methylation levels between 100 days and one-year follow-up. Hence, the effects on gene expression occurred predominantly during treatment, while the effects on DNA methylation occurred after treatment. In conclusion, unrecognized side effects of long-term amoxicillin treatment were revealed, as alterations were observed in both gene expression and DNA methylation that lasted long after the end of treatment.

## 1. Introduction

Amoxicillin is a broad-spectrum penicillin antibiotic that acts by inhibiting the synthesis of the peptidoglycan layer of bacterial cell walls. The cell walls are weakened, leading to lysis of the cell. Amoxicillin is used to treat a variety of bacterial infections, e.g., in the lungs, tonsils, mouth, ears, nose, throat, skin, urinary tract, and stomach [1]. It is one of the most commonly prescribed antibiotics worldwide and is listed by the WHO as one of the essential antibiotics for children [2]. Long-term treatment with amoxicillin is prescribed for some conditions, such as Lyme and Actinomycosis disease, prosthetic joint infection, post-splenectomy, and anthrax prophylaxis [3]. Most reports on the drug safety of amoxicillin are based on short-term use, and little is known about potential long-term adverse effects.

Lately, amoxicillin has been suggested as a possible treatment strategy for subgroups of low back pain (LBP) patients. LBP, the leading cause of disability globally [4], causes activity limitation and work absence with large personal and societal consequences [5]. However, most reported cases of LBP do not have clear anatomical explanations, and current treatment options are few and have small or negligible effects [5]. A subgroup of LBP patients has Modic changes (MC), vertebral bone marrow changes detected by magnetic resonance imaging [6,7]. The pathobiology of the MCs and their biological connection to LBP are uncertain [8]; however, one hypothesis is that LBP and MC are caused by infection of an intervertebral disc by the *Cutibacterium acnes* bacteria, triggering a detrimental cascade of inflammation in the vertebrae [9]. Long-term antibiotic treatment has therefore been proposed as a possible treatment strategy for these patients, and two randomized controlled trials have assessed the efficacy of 100 days of amoxicillin treatment. Whereas Albert et al. [6] reported a substantial effect, Bråten et al. [7] did not replicate any clinically relevant effect of amoxicillin treatment compared with placebo. Hence, the two trials have revealed conflicting results, which could have important implications for LBP treatment guidelines.

In addition to traditional pharmaceutical effects, there is increasing evidence that common medications such as antidepressants, antidiabetic drugs, opioids, cannabinoids, anti-seizure medication, cytostatics, and paracetamol may influence gene expression and epigenetic patterns [10,11,12,13,14,15,16]. It is not known whether long-term use of antibiotics has epigenetic side effects that potentially influence the regulation of gene expression through direct or indirect mechanisms. Secondary epigenetic and transcriptomic effects associated with antibiotic treatment may persist after it is discontinued. If so, it could have implications for and challenge the current understanding of long-term antibiotic treatment.

In this study, we have investigated whether 100 days of amoxicillin treatment are associated with gene expression and DNA methylation changes in whole blood from patients enrolled in the Antibiotics in Modic Changes (AIM) study [7]. To our knowledge, this is the first study investigating the effect of long-term antibiotic treatment on gene expression and epigenetic outcomes.

## 2. Results

### 2.1. Study Cohort Characteristics

We included all eligible patients with blood sampling at three time points receiving amoxicillin or placebo in the AIM study in the analyses (*n* = 100). The gene expression data included 84 patients with MC type I (edema type) who either received amoxicillin (*n* = 44) or placebo (*n* = 40) treatment. The DNA methylation data included an additional 16 patients with MC type II (fatty type) but not MC type I, for a total of 100 patients (*n* = 52 and *n* = 48 patients receiving amoxicillin and placebo treatment, respectively). The patients in both treatment groups were similar with regards to gender, age, back pain severity, and other clinical characteristics measured (Table 1).

### 2.2. Long-Term Amoxicillin Treatment Did Not Alter Blood Cell-Type Composition

The proportions of the 12 leukocyte cell types estimated from the DNA methylation data were within normal clinical ranges and showed expected variation among the patients (Appendix A). The cell-type proportions did not show any significant change over time in either the amoxicillin or placebo groups (Appendix A). Hence, this suggests that treatment with amoxicillin and the underlying chronic, inflammatory LBP condition did not alter the cell-type composition, and we did not consider this a confounding source of variability in the downstream gene expression and DNA methylation analyses.

### 2.3. Altered Gene Expression in Amoxicillin Treatment Group

In the group of patients receiving amoxicillin, we found 28 genes significantly differentially expressed between screening and 100 days (false discovery rate (FDR) < 0.05); three genes were upregulated, and 25 genes were downregulated (Table 2 and Table 3A). From 100 days to one-year follow-up, seven genes were differentially expressed, including five of the significant genes from the first interval (*GLDC*, *IGHV366*, *IGKV19*, *IGKV3D15,* and *IGLV327*) (Table 2 and Table 3B, and Appendix A). No significant changes in gene expression were found between screening and one-year follow-up. Remarkably, among the 30 genes showing differential gene expression in at least one time interval, 21 were immunoglobulin (Ig) variable chain genes (Table 3).

We observed a general trend in the gene expression levels in the patients receiving amoxicillin treatment, where the direction of change between screening and 100 days was opposite of the changes between 100 days and one year for 73% of all genes (Figure 1A). In particular, all the significant genes followed this pattern. 62% of the genes following this trend did not reach their original expression value measured at screening. We did not observe this trend in the placebo group (Figure 1B).

Only two genes were differentially expressed between screening and 100 days in the placebo group (Table 3C). One of these (*HBG1*) was a hemoglobin-coding gene that erroneously appeared differentially expressed because of incomplete globin depletion in a few samples. The second gene (*PROX1*), coding for a homeobox transcription factor, was significantly downregulated between screening and 100 days. No genes were significantly differentially expressed between 100 days and one year, or screening and one year, in the placebo group.

We controlled for a slight inflation in t-statistics using the Bayesian method implemented in the bacon package (Appendix A).

### 2.4. Altered DNA Methylation in Amoxicillin Treatment Group

Paired analyses between time points revealed widespread alterations in DNA methylation in the amoxicillin group (Figure 1C). While we did not observe any significant differences from screening to 100 days, one CpG was differentially methylated from screening to one year, and 4547 CpGs annotated to 2844 genes were differentially methylated from 100 days to one year (Table 2 and Appendix A). Of the 4547 differentially methylated CpGs, 97.6% (*n* = 4442 CpGs) showed an increase in DNA methylation and 2.4% a decrease in DNA methylation (*n* = 105) from 100 days to one year. Overall, the effect sizes were small, and there was no enrichment of gene ontology terms or specific pathways. The results corroborate the trend observed in the gene expression data, as a small change in DNA methylation in the opposite direction was observed from screening to 100 days for these CpGs. Thus, we did not identify differential gene expression or DNA methylation across the whole intervention period, from screening to one year. However, while effects on gene expression occurred during treatment, effects on DNA methylation were seen after treatment. Complementing the findings from the gene expression analysis, we did not observe any significant changes in DNA methylation in the placebo group (Figure 1D). These results indicate that the methylation of CpGs on the EPIC array is longitudinally stable over a period of one year.

Quantile-quantile plots of the observed versus expected *p*-values showed inflation in the t-statistics between 100 days and one year (Appendix A). However, surrogate variable analysis did not detect any hidden covariates with a global effect on DNA methylation associated with amoxicillin exposure between the time points.

### 2.5. Low Overlap of Genes in the Gene Expression and DNA Methylation Data

Not all genes are covered by the EPIC array, as measurement of DNA methylation can be problematic in polymorphic and repetitive genomic regions [17]. Therefore, to investigate the interplay between the observed gene expression and DNA methylation changes, we first checked to what extent the data sets contained information on the same genes. Using the UCSC Refgene Name composite annotation track, we identified 14,813 genes with both gene expression and DNA methylation successfully measured. Most of the expressed protein-coding genes were also annotated as CpGs in the DNA methylation data set. Some gene categories were completely absent in the methylation data (Figure 2), including the Ig genes that are not covered on the EPIC array. Among the differentially expressed genes, only 8 out of 32 genes were covered in the DNA methylation dataset (i.e., *SERPINB6*, *HRASLS2*, *CIAPIN1*, *GLDC*, *DACT1*, *LINC01093*, *PROX1*, and *HBG1*). One of these, *SERPINB6*, was also differentially methylated (cg09080894) from 100 days to one year in the amoxicillin group. The incomplete overlap between the datasets, particularly the lack of coverage of Ig genes in the DNA methylation dataset, limited further integrative analyses of significant genes from the individual analyses.

However, explorative analyses of the overall trend in correlation of changes in gene expression and DNA methylation for the overlapping data are shown in Figure 3 (*n* = 461,109 pairs). Of note, the gene expression data were represented by several and differing numbers of DNA methylation sites (probes), which is evident from the vertical lines in the plot. In the placebo group, the direction of changes in gene expression and DNA methylation is quite evenly distributed across all time points, and we observe no general trend between the gene expression and DNA methylation changes. In the amoxicillin group, there was a moderate trend of negative correlations with opposite directions between time points (the lower right quadrant of Figure 3A and the upper left quadrant of Figure 3C). From screening to 100 days, more genes have upregulated expression and decreased DNA methylation (low-right and top-left quadrants in Figure 3A), whereas from 100 days to one year, the trend is the opposite, with more genes being downregulated and more methylated (Figure 3C).

## 3. Discussion

To our knowledge, this is the first study investigating the molecular effects of long-term treatment with amoxicillin. We found significant alterations in gene expression and DNA methylation in patients receiving amoxicillin, which were not observed in patients receiving placebo. The changes were still present at one-year follow-up, nine months after the end of treatment, and represent potential side effects of long-term antibiotic treatment.

Certain conditions and medications may lead to altered white blood cell counts [18]. It is well known that subgroups of white blood cells display distinct gene expression and DNA methylation profiles and may represent a potential confounding factor in the case of a change in cell subgroup composition [19,20]. Interestingly, amoxicillin treatment did not result in any significant alterations in the estimated cell type proportions of 12 leukocytes in our study, and therefore cell type composition was not corrected for in our models. There might be subtle cell-type-specific effects that are masked or only recognizable in combinations of cell types, but which would require single-cell sequencing technologies to be explored.

We observed a global trend of changes in gene expression in patients treated with amoxicillin. The majority of gene expression changes observed after 100 days of treatment were subsequently reversed at one-year follow-up; however, a large part (63%) of these genes did not fully return to the original expression values measured at screening. This trend was not observed in the placebo group, where the changes in expression were evenly distributed in both directions at both 100 days and one-year follow-up and might therefore represent a general deregulation of gene expression during amoxicillin treatment that is not yet restored at one-year follow-up. Sustained gene expression changes observable 6 months after antibiotic treatment have been reported in Lyme disease patients [21]. Although this might originate from the shared disease history of the patients, the altered gene expression was similar regardless of whether the patients had persistent symptoms or not. The antibiotic treatment could therefore be the cause of the prolonged transcriptome deregulation observed. Furthermore, deregulation (predominantly reduction) in gene expression as an effect of antibiotic intake has been observed in mice [22], attributed to the combined effects of a reduction in microbiota, the effects of remaining antibiotic resistant microbes, and the direct effects of antibiotics on the host tissues.

Significant gene expression changes were primarily observed between the start and end of amoxicillin treatment, and 25 of 28 genes were downregulated. At one-year follow-up, all these genes were reversed towards their original expression; however, only five were reversed significantly. Of particular interest, 19 of the genes significantly downregulated encode Igs. Ig genes encode either B cell surface receptors triggering B cell activation upon antigen binding or secreted antibodies, which are key initiators of a range of downstream immune responses. A large number of genes encode the two heavy and two light chains, i.e., kappa or lambda, making up the Igs. The differentially expressed Ig genes in our study are all coding for Ig variable domains, which are the sites for antigen recognition on either heavy or light Ig chains. A reduction in the expression of these genes could therefore potentially affect a patient’s immune response to succeeding infections. To further explore the role of amoxicillin on the expression of the Ig genes, a detailed investigation of the regulatory mechanisms behind the observed downregulation could be helpful. The Ig gene segments are extremely polymorphic, and in-depth analyses require specific tools able to distinguish between the gene variants at higher resolution [23]. The high degree of polymorphism is probably a challenge for the design of unique probes to map DNA methylation patterns at Igs and is likely the reason why the genes are not represented on the EPIC methylation array. In contrast, we interrogated gene expression by sequencing, which is a technology not to the same extent limited by polymorphisms.

The use of antibiotics is known to have long-lasting effects on the host’s gut microbiota. Loss of bacterial diversity, including certain central species, leads to reduced colonization resistance against invading pathogens and advanced antibiotic resistance, as well as changes in host metabolism [24]. Both depletion of the microbiota and amoxicillin treatment have been shown in mice to reduce the levels of Igs (of type A) in the blood, with a subsequent influence on increased susceptibility to future infections [25]. As effects on the gut microbiota can be observed even after short-term amoxicillin use [26], it is likely that long-term use will have even more profound consequences. In fact, a previous analysis of the microbiota in a small portion of AIM patients revealed a reduction in species diversity and a shift in overall microbiome composition in the amoxicillin-treated patients (*n* = 8), while the placebo patients were stable over time (*n* = 12) [27]. The downregulation of Igs observed in our antibiotic-treated patients could therefore be related to a disturbed microbiota, with a smaller variety of bacterial species present putting less pressure on the adaptive immune system.

Treatment with amoxicillin was also associated with genome-wide longitudinal DNA methylation changes, which were evident even after one year. Overall, the effect sizes were small, with a predominant increase in DNA methylation distributed across many genes. This increase in DNA methylation was not observed across the whole intervention period, from screening to one-year follow-up. In fact, the majority of CpGs showed a slight decrease in DNA methylation from screening to 100 days before the subsequent larger increase from 100 days to one year. Hence, such changes were not detected in our linear models. There was no enrichment of gene ontologies and specific pathways, which suggests a widespread influence with long-term effects on DNA methylation outcomes in peripheral blood in these patients. Unfortunately, the poor coverage of Igs on the EPIC arrays limits the identification of a similar effect on DNA methylation and the investigation of a regulatory role of DNA methylation on the expression of these genes in our data.

In contrast to the Ig genes, most protein-coding genes (97%) were included in both datasets; however, only one gene showed significant changes in both methylation and gene expression. The lack of overlap and the fact that gene expression changes appear to occur during amoxicillin treatment while methylation changes occur later could indicate that multiple regulatory mechanisms have been affected. The early gene expression alterations, particularly targeting Ig genes, could either be regulated by non-methylation-driven transcription factors or undetected methylation changes in their regulatory region [28]. The DNA methylation changes detected post-treatment could potentially also produce rise to subsequent alterations in gene expression occurring after our one-year observation period.

In the placebo group, very few changes were observed overall. This demonstrates that DNA methylation and gene expression are longitudinally stable during the intervention period of one year and supports the fact that our findings are truly related to the influence of amoxicillin exposure. This is in line with other studies showing that DNA methylation is stable and, to a low extent, explains short-term changes in gene expression in clinical trials [29,30]. Our results thus provide insight into transcriptomic and epigenetic dynamics, with implications for the interpretation of findings in other clinical trial studies.

This study has strengths and limitations. We observed inflation in the t-statistics between 100 days and one year in the DNA methylation data, which could not be fully corrected for using bacon. A systematic analysis of possible covariates known to be associated with either DNA methylation and/or patient characteristics did not reveal any significant association in the DNA methylation data or within patients across time points. As we cannot exclude potential unmeasured confounding, future studies are needed to replicate the results presented in this study in other patient populations. Nevertheless, the patients were randomized during the intervention and throughout the generation and preprocessing of the gene expression and DNA methylation data. Therefore, technical and biological confounding of the data is minimized and unlikely to influence specific time points in only the patients receiving amoxicillin treatment.

Another important consideration is that long-term antibiotic treatment is prescribed to patients of all ages and with a variety of conditions [3]. It is therefore vital that the side effects of such treatment are studied across several patient groups and antibiotic types. The patients were included in a clinical study of patients with chronic LBP and MC, and future studies should investigate gene expression and DNA methylation outcomes associated with other conditions. Nevertheless, since LBP is quite common among adults [31], we believe that the results presented here have broad relevance with potential implications for other patient groups prescribed long-term amoxicillin. Furthermore, while our gene expression analysis included patients with only MC type I, the DNA methylation analysis additionally included data from 16 patients with MC type II. However, as MC types I and II are regarded as interconvertible and do not appear to represent clinical differences [32], these patients are likely to respond similarly to amoxicillin treatment in terms of gene expression or DNA methylation outcomes. Additionally, other tissues, such as saliva, should be explored to complete the picture of potential side effects of the treatment [33].

Lastly, we have not investigated whether the identified changes in gene expression or DNA methylation have any prognostic or therapeutic implications or whether they are related to changes in the patients’ clinical status. However, the patients in the two treatment groups had comparable demographic and clinical characteristics at screening, and the clinical trial did not report clinically relevant changes between the treatment groups [7], which suggests that our findings are related to and reflect amoxicillin side effects.

## 4. Materials and Methods

### 4.1. Study Cohort

The study cohort is a sub-sample of patients enrolled in the AIM study, which is a double-blinded, randomized, placebo-controlled, multicenter trial assessing the efficacy of 100 days of amoxicillin treatment in patients with chronic LBP and MC [7]. The patients suffered from substantial LBP (with a mean intensity of ≥5 on three 0–10 numerical rating scales at the time of inclusion) and had little other comorbidity. Eligibility criteria and the study protocol for the AIM study are fully published elsewhere [34]. Patients of Caucasian ethnicity and with successful blood sampling at three time points (screening, after 100 days of treatment, and at one-year follow-up) were included in this study.

### 4.2. Isolation and Preparation of RNA and DNA Samples

Peripheral blood for gene expression and DNA methylation analyses was sampled simultaneously in Tempus Blood RNA Tubes (Thermo Fisher Scientific, Waltham, MA, USA) and K2-EDTA tubes (Becton, Dickinson, and Company, Franklin Lakes, NJ, USA), respectively. Total RNA was isolated from the Tempus Blood RNA Tubes using the Preserved Blood RNA Purification Kit I (Norgen Biotek, Thorold, ON, Canada) according to the manufacturer’s instructions. DNAse treatment was carried out as recommended. The quality and concentration of the RNA were measured using the BioAnalyzer 6000 Nano kit (Agilent Technologies, Santa Clara, CA, USA) and Qubit RNA HS (Thermo Fisher Scientific), with a mean RNA integrity number (RIN) of 9.2 and a concentration of 159 ng/μL. The total RNA samples were depleted for ribosomal RNA and globin transcripts with the Globin-Zero^®^ Gold rRNA Removal Kit (Illumina, San Diego, CA, USA).

DNA was isolated from the K2-EDTA tubes using the Maxwell 16 Cell LEV DNA Purification Kit (Promega, Madison, WI, USA) according to the manufacturer’s instructions. The quality and concentration of the DNA were measured using Nanodrop (Thermo Fisher Scientific). Bisulfite conversion of DNA (500 ng) was done using the EZ-96 DNA Methylation-Gold Kit (Zymo Research, Irvine, CA, USA) according to the manufacturer’s instructions.

### 4.3. Generation, Preprocessing and Quality Control of Data

RNA samples for sequencing were prepped using TruSeq RNA library prep kits (Illumina) and sequenced with a 2 × 75 bp paired-end configuration on the HiSeq3000 platform (Illumina). The quality of the sequencing was assessed using FastQC and Qualimap [35,36]. Preprocessing and quality control of the RNA sequencing data are described in detail elsewhere [37]. Briefly, 97.4% of the reads were successfully mapped to the human genome (GRCh38.p10) using HISAT2 v2.1.0 [38], and 45.3% of these were assigned to genes with feature counts from Subread v1.6.3 [39], using gene coordinates from Ensembl 88 [40]. Only autosomal genes were kept for further analyses, and lowly expressed genes were filtered out using the *filterByExpr* function (min.count = 1) in the R package edgeR [41], ultimately leaving 21,835 and 21,890 genes for downstream analysis in the amoxicillin and placebo groups, respectively. The raw counts were normalized using edgeR’s trimmed mean of M-values method.

DNA methylation was measured using the Infinium MethylationEPIC BeadChip array (Illumina) according to the manufacturer’s instructions at the Institute of Life and Brain Sciences at the University of Bonn, Germany. Normalization of the measurements was performed with the Beta-Mixture Quantile (BMIQ) procedure [42], background subtraction was performed using *oob.enmix* [43], and further preprocessing and quality control were done using the R package RnBeads v.2.4.0 [44]. Specifically, cross-reactive probes [45,46] (*n* = 43,256), probes with overlapping SNPs in any of the bases in the target sequence (*n* = 41,930), and probes with unreliable measurements (detection *p* values > 0.01) (*n* = 14,576 probes) were removed. After further filtering out sex-chromosomal probes (*n* = 18,814 probes) and non-CpG probes (*n* = 2458), a total of 775,684 probes were included in the final data set.

Principal component analysis (PCA) did not identify any batch effects in either data set (Appendix A). Further, as a check for sample swaps, DNA methylation at 59 control probes on the EPIC array was plotted in a heatmap including all 300 samples (Appendix A). The plot shows correct intra-individual clustering for all 100 individuals.

### 4.4. Cell Deconvolution

This study is based on samples from whole blood, which is a complex tissue comprising many different leucocyte cell types. Leukocyte cell types display very different gene expression and DNA methylation profiles, and variation in blood cell composition may confound downstream analyses [19,20,47]. As both treatment with amoxicillin and the underlying chronic, inflammatory LBP condition can alter the cell-type composition, we performed cell deconvolution and estimated 12 immune cell populations (i.e., neutrophils, eosinophils, basophils, monocytes, natural killer cells, regulatory T cells, naive and memory B cells, CD4^+^ and CD8^+^ T cells) before and after treatment from the DNA methylation data using FlowSorted.BloodExtended.EPIC R library [20,48]. A paired t-test was used to test whether the cell type estimates changed significantly over time.

### 4.5. Identification of Covariates and Batch Effects

PCA was used to explore potential batch effects. Specifically, we analyzed the association of categorical and continuous covariates with the principal components. Paired *t*-tests were performed to test whether potential covariates significantly associated with PCs changed over time in both groups.

### 4.6. Differential Gene Expression and DNA Methylation Analysis

Longitudinal intra-individual differences in gene expression and DNA methylation levels were tested separately in each treatment group (a) between screening and 100 days, (b) between screening and one-year follow-up, and (c) between 100 days and one-year follow-up. The statistical analyses were performed with the R package limma v3.48.3 [49,50]. The data sets were log-transformed prior to differential analyses. Specifically, the voom function [51] was used to transform the expression read counts to log2 counts per million reads with associated weights, while methylation beta values were logit transformed to M values. Identification of differential gene expression was done by fitting a linear regression model implemented in limma to each gene. The *p*-value distributions were estimated empirically from the t-scores using the Bayesian method implemented in the bacon v1.26.0 R package [52] to correct for a slight inflation in the test statistics. The differential DNA methylation analysis was performed using the same linear regression model on the M values. Annotation of CpG sites was done with the package IlluminaHumanMethylationEPICanno.ilm10b4.hg19 [53]. The FDR was controlled at 0.05 using the Benjamini-Hochberg method in both analyses [54]. We performed gene ontology analysis on the significantly differentially methylated DNA sites using the *gometh* function implemented in the MissMethyl package [55], which accounts for differences in the number of probes on the EPIC array annotated to the genes as well as multi-gene annotations. All plots were generated using the ggplot2 v3.4.2 R package [56].

### 4.7. Integration of Gene Expression and DNA Methylation Data

Integrative analysis of changes in expression and DNA methylation was done for genes measured in the gene expression dataset that overlap with CpGs in the DNA methylation dataset. The CpGs were annotated using the UCSC Refgene Genes composite track annotation in the *IlluminaHumanMethylationEPICanno.ilm10b4.hg19* package.

## 5. Conclusions

This study shows that amoxicillin treatment in patients with LBP and MC is associated with transcriptomic and epigenetic changes observable long after the end of treatment. The results from this study may potentially have implications for medication guidelines for any patient group receiving long-term amoxicillin treatment. Future studies investigating both microbiota and single-cell transcriptomic and epigenomic analyses, as well as targeted DNA methylation analysis of Ig genes, are needed to improve interpretation and unravel the complex interplay between antibiotics, microbiota, and immunity.

## Figures and Tables

**Figure 1 antibiotics-12-01217-f001:**
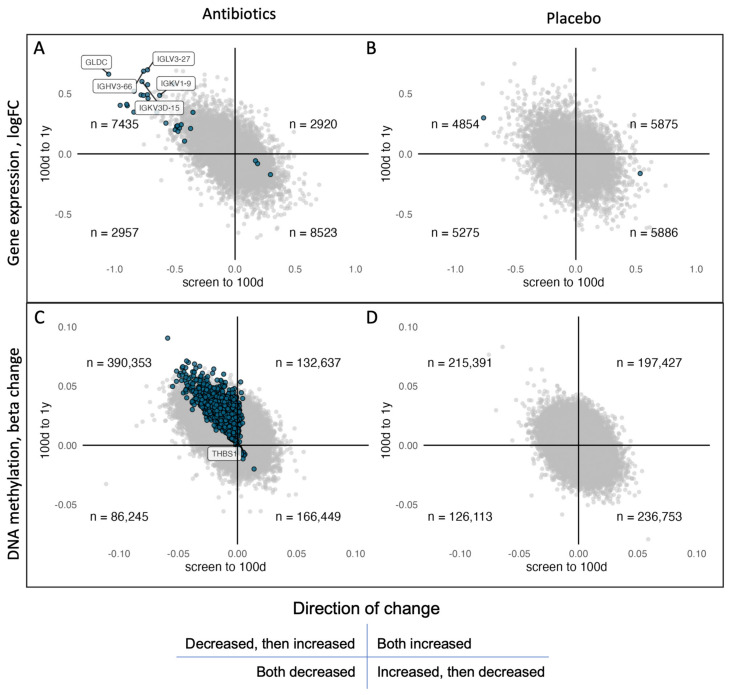
Changes in gene expression and DNA methylation in the two treatment groups from screening to 100 days (x-axes) versus 100 days to one year (y-axes). (**A**,**B**) Log fold changes in gene expression of all expressed genes. (**C**,**D**) Changes in methylation beta value at all CpG sites. The number of genes/sites in each quadrant is shown. Significant (in either interval) genes/sites are colored, and those significant in both intervals are labeled with gene names.

**Figure 2 antibiotics-12-01217-f002:**
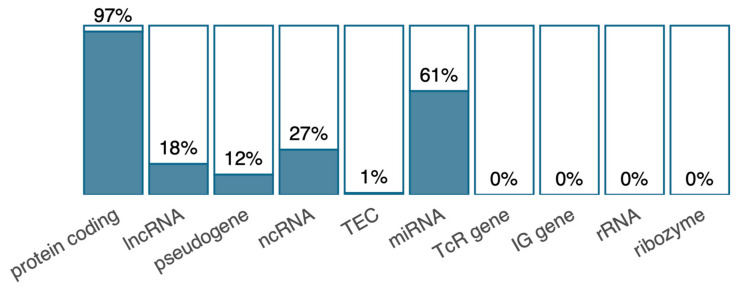
Overlapping genes in gene expression and DNA methylation data sets. The bars show the percentage of expressed genes with at least one annotated CpGs for different gene biotypes (categories).

**Figure 3 antibiotics-12-01217-f003:**
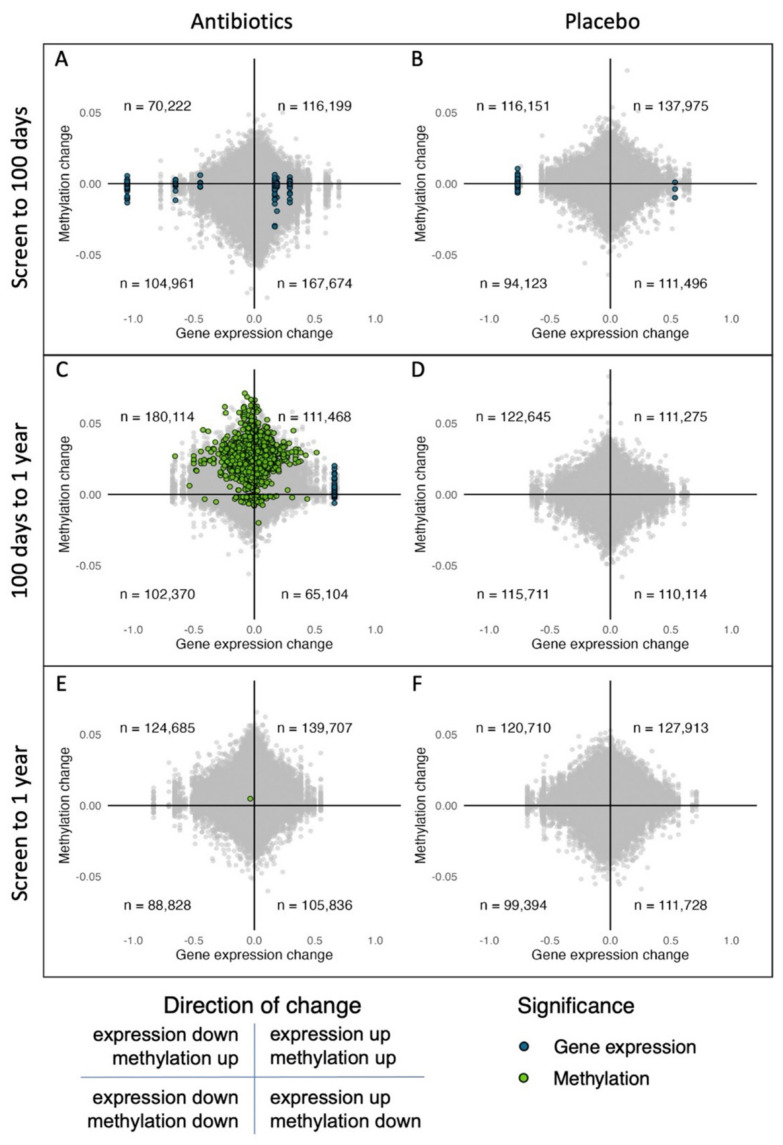
Change in DNA methylation vs. change in gene expression between time points. Each dot represents a CpG site and its annotated gene. (**A**) Screening to 100 days in the amoxicillin group; (**B**) Screening to 100 days in the placebo group; (**C**) 100 days to one year in the amoxicillin group; (**D**) 100 days to one year in the placebo group; (**E**) Screening to one year in the amoxicillin group; and (**F**) Screening to one year in the placebo group. Significant changes identified in the separate analyses above are colored (blue = gene expression analyses, and green = DNA methylation analyses). The numbers of overlapping gene sites in each quadrant are shown.

**Table 1 antibiotics-12-01217-t001:** Demographic and clinical characteristics of patients (*n* = 100) at screening. There were no significant differences between the groups.

	Treatment Group	
	Placebo (*n* = 48)	Amoxicillin (*n* = 52)
Female	60%	63%
Age (mean, SD)	45.7 (8.9)	45.9 (7.6)
BMI (median ^†^, IQR)	24.6 (5.1)	25.4 (5.2)
Smoking, *n* = 98	20%	31%
Disability, RMDQ (mean, SD), *n* = 93	12.5 (3.8)	13.2 (4.9)
LBP intensity, NRS (mean, SD), *n* = 99	6.6 (1.2)	6.6 (1.1)
Previously operated for disc herniation	33%	29%
LBP duration in years (mean, SD), *n* = 99	6.3 (5.5)	4.9 (5.2)
Glucose, mmol/L (mean, SD), *n* = 84	5.10 (0.6)	5.10 (0.7)
Thrombocytes, ×10^9^/L (mean, SD)	259 (78)	268 (53)
Haemoglobin, g/100 mL (mean, SD), *n* = 99	14.2 (1.2)	14.2 (1.2)
Hematocrit, % (mean, SD)	40 (4)	40 (3)
Creatinine, µmol/L (mean, SD)	70.8 (14.6)	69.8 (12.8)
ASAT, U/L (mean, SD), *n* = 99	22.7 (5.3)	24.5 (7.8)
CRP, mg/L (mean, SD), *n* = 99	1.5 (1.9)	2.0 (4.0)
WBC, ×10^9^/L (mean, SD)	6.5 (1.8)	6.5 (1.7)

ASAT: Aspartate aminotransferase; BMI: Body mass index; CRP: C-reactive protein; IQR: Interquartile Range; LBP: Low back pain; NRS: Numerical rating scale; RMDQ: Roland-Morris Disability Questionnaire; SD: Standard deviation; WBC: White blood cell count. ^†^ Median shown instead of mean because of skewed distribution of BMI in the population.

**Table 2 antibiotics-12-01217-t002:** Overview of significant changes in gene expression and DNA methylation.

	Gene Expression		DNA Methylation	
Time Interval	Amoxicillin	Placebo	Amoxicillin	Placebo
screening—100 d	↑ 3	↑ 1 *		
	↓ 25	↓ 1		
100 d—1 y	↑ 7		↑ 4442	
			↓ 105	
screening—1 y			↑ 1	

↑: Increased gene expression/increased methylation of CpG. ↓: Decreased gene expression/decreased methylation of CpG. * Spurious finding.

**Table 3 antibiotics-12-01217-t003:** Significantly differentially expressed genes in patients on antibiotics (*n* = 44), from screening to 100 days (**A**) and 100 days to one year (**B**), and patients on placebo (*n* = 40), from screening to 100 days (**C**). No genes were significant in patients on placebo from 100 days to one year.

(**A**) Antibiotic group, screening—100 days			
Ensembl ID	Gene name	LogFC	Padj
ENSG00000178445	*GLDC*	−1.05	7.6 × 10^−6^
ENSG00000211669	*IGLV3-10*	−0.896	1.7 × 10^−4^
ENSG00000211964	*IGHV3-48*	−0.575	1.7 × 10^−4^
ENSG00000133328	*HRASLS2*	−0.651	2.5 × 10^−4^
ENSG00000242766	*IGKV1 D-17*	−0.955	3.6 × 10^−4^
ENSG00000224041	*IGKV3D-15*	−0.774	1.6 × 10^−3^
ENSG00000165617	*DACT1*	0.294	3.3 × 10^−3^
ENSG00000211662	*IGLV3-21*	−0.781	2.8 × 10^−3^
ENSG00000211937	*IGHV2-5*	−0.497	4.0 × 10^−3^
ENSG00000249173	*LINC01093*	−0.447	4.2 × 10^−3^
ENSG00000211611	*IGKV6-21*	−0.901	4.2 × 10^−3^
ENSG00000251546	*IGKV1D-39*	−0.839	4.2 × 10^−3^
ENSG00000211625	*IGKV3D-20*	−0.729	6.0 × 10^−3^
ENSG00000241755	*IGKV1-9*	−0.627	6.0 × 10^−3^
ENSG00000233030	*RP11-196G18.3*	−0.462	8.2 × 10^−3^
ENSG00000005194	*CIAPIN1*	0.187	2.1 × 10^−2^
ENSG00000211942	*IGHV3-13*	−0.474	1.4 × 10^−2^
ENSG00000211941	*IGHV3-11*	−0.42	1.9 × 10^−2^
ENSG00000124570	*SERPINB6*	0.169	3.0 × 10^−2^
ENSG00000211655	*IGLV1-36*	−0.761	2.3 × 10^−2^
ENSG00000224220	*AC104699.1*	−0.727	2.9 × 10^−2^
ENSG00000211659	*IGLV3-25*	−0.724	3.2 × 10^−2^
ENSG00000211658	*IGLV3-27*	−0.728	3.3 × 10^−2^
ENSG00000211972	*IGHV3-66*	−0.76	4.3 × 10^−2^
ENSG00000211943	*IGHV3-15*	−0.369	4.5 × 10^−2^
ENSG00000243290	*IGKV1-12*	−0.485	4.5 × 10^−2^
ENSG00000211663	*IGLV3-19*	−0.479	4.6 × 10^−2^
ENSG00000230709	*AC104024.1*	−0.841	4.8 × 10^−2^
(**B**) Antibiotic group, 100 days—one year			
Ensembl ID	Gene name	LogFC	Padj
ENSG00000276566	*IGKV1D-13*	0.577	7.2 × 10^−3^
ENSG00000178445	*GLDC*	0.662	1.0 × 10^−2^
ENSG00000211658	*IGLV3-27*	0.698	1.0 × 10^−2^
ENSG00000224041	*IGKV3D-15*	0.601	1.0 × 10^−2^
ENSG00000211972	*IGHV3-66*	0.686	3.5 × 10^−2^
ENSG00000232216	*IGHV3-43*	0.345	3.5 × 10^−2^
ENSG00000241755	*IGKV1-9*	0.486	4.2 × 10^−2^
(**C**) Placebo group, screening—100 days			
Ensembl ID	Gene name	LogFC	Padj
ENSG00000213934	*HBG1*	0.534	1.5 × 10^−3^
ENSG00000117707	*PROX1*	−0.766	3.3 × 10^−3^

LogFC: Log fold change; Padj: adjusted *p*-value.

## Data Availability

The data presented in this study are available on request from the corresponding author.

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
