# Peer review of "Long-Term Use of Amoxicillin Is Associated with Changes in Gene Expression and DNA Methylation in Patients with Low Back Pain and Modic Changes"

_antibiotics, 2023, doi:10.3390/antibiotics12071217_

Round 1
Reviewer 1 Report
This study demostrates that changes in gene expression and DNA methylation pattern occur in blood cells due to long-term treatment with the antibiotic amoxicillin.
I have some comments:
Table 1: a) CRP protein increases in inflammatory processes, In clinical tests, its quantitation is expressed as mg/dl or mg/L. Why you use U/L? b)The hematocrit % says 0.4; is it 40?? Please explain or fix it.
There is a recurring mistake in several pages, it says: Error, Reference not found. Please fix it.
Could you discuss a little about what would happen in other tissues, since the antibiotics has systemic affects?
Similar studies have been perform on culture cells, for example Ryu et al (2017) "Antibiotics affect gene expression in culture cells" Sci. Rep 7, 7533.
It would be possible to perform a similar study (in vivo) on mucosal cells which can be easy to obtain?
In my opinion this draft contributes with a study method to better understand the side effects of antibiotics or any drug.
Reviewer 2 Report
The article entitled "Long-term antibiotic use is associated with changes in gene expression and DNA methylation" provides a concise but impactful exploration of the potential consequences of prolonged antibiotic treatment on epigenetic regulation of the genome.
The topic is very interesting and poorly explored in the literature, so I believe this study can offer a useful contribution. The article is well written, in particular the methods and results are clear, the discussion detailed, and the conclusions consistent with the results.
I have only a couple of requests for the authors. Since the study was conducted on patients undergoing antibiotic treatment for the same condition, namely low back pain with modic changes, I think this condition deserves more space. Therefore, I would like to ask the authors to include a reference to the pathology in the title and to elaborate on its description in the introduction.
Thank you.
Reviewer 3 Report
- I suggest that the title could be changed to: "Long-term use of amoxicillin is associated with changes in gene expression and DNA methylation" rather than generalizing to all antibiotics.
- In some parts of the text, the message "Error! Reference source not found." appear.
- The relationship between changes in gene expression and DNA methylation and LBP and MC is not clear in the discussion and conclusion regarding the consequences of this association.
- In line 212, the sentence "The consequences of these alterations are not known but should be considered when exploring potential side effects of long-term antibiotic treatment" is not very clear. Perhaps the authors should describe the possible consequences of these changes and explain why they are important.
Round 2
Reviewer 3 Report
The paper is suitable for publication in present form.